# Learning to Imitate with Less: Efficient Individual Behavior Modeling in Chess

**Zhenwei Tang**        *josephtang@cs.toronto.edu*
*University of Toronto*

**Difan Jiao**        *difanjiao@cs.toronto.edu*
*University of Toronto*

**Eric Xue**        *e.xue@mail.utoronto.ca*
*University of Toronto*

**Reid McIlroy-Young**        *reidmcy@seas.harvard.edu*
*Harvard University*

**Jon Kleinberg**        *kleinberg@cornell.edu*
*Cornell University*

**Siddhartha Sen**        *sidsen@microsoft.com*
*Microsoft Research*

**Ashton Anderson**        *ashton@cs.toronto.edu*
*University of Toronto*

**Reviewed on OpenReview:** *https://openreview.net/forum?id=iw4kjcw319*

## Abstract

As humans seek to collaborate with, learn from, and better understand artificial intelligence systems, developing AIs that can accurately emulate individual decision-making becomes increasingly important. Chess, a long-standing AI benchmark with precise skill measurement, offers an ideal testbed for human-AI alignment. However, existing approaches to modeling human behavior require prohibitively large amounts of data from each individual, making them impractical for new or sparsely represented users. In this work, we introduce Maia4All, a framework designed to learn and adapt to individual decision-making styles efficiently, even with limited data. Maia4All achieves this through a two-stage optimization process: (1) an *enrichment* step, which bridges population and individual-level human behavior modeling with a prototype-enriched model, and (2) a *democratization* step, which leverages ability levels or user prototypes to initialize and refine individual embeddings with minimal data. Our experimental results show that Maia4All can accurately predict individual moves and profile behavioral patterns with high fidelity, establishing a new standard for personalized human-like AI behavior modeling in chess. Maia4All achieves individual human behavior modeling in chess with only 20 games, compared to the 5,000 games required previously, representing a significant improvement in data efficiency. Our work provides an example of how population AI systems can flexibly adapt to individual users using a prototype-enriched model as a bridge. This approach extends beyond chess, as shown in our case study on idiosyncratic LLMs, highlighting its potential for broader applications in personalized AI adaptation. Maia4All implementation is available at `https://github.com/CSSLab/maia4all`.

# 1 Introduction

The rise of artificial intelligence (AI) systems that rival or surpass human ability in domains where people remain active has introduced the possibility of collaborating with and learning from AI agents. A line of research has pursued this vision in the model system of chess, in which AI became superhuman 20 years ago, people vary widely in their ability, and vast detailed datasets of action traces abound. Since capturing human decision-making style is a prerequisite to algorithmically-informed teaching and collaboration, previous work has focused on creating AI agents that mimic human play (McIlroy-Young et al., 2020; 2022; Jacob et al., 2022; Tang et al., 2024). Further, since capturing *individual* decision-making style is a prerequisite to personally tailored algorithmic instruction, researchers have developed models of how specific people play chess, surpassing population models in their accuracy rates on their target individual's decisions (McIlroy-Young et al., 2022).

However, these fine-tuning-based models of individual decision-making require extraordinary amounts of data per person to function. When Maia, a human-like chess engine, was fine-tuned to play like specific individuals, gains in accuracy over base Maia were only achieved when the player had 5,000 games worth of data (McIlroy-Young et al., 2022). This is an immense amount of game-playing; a typical person would take around 1,000 hours to play this many games, which is equivalent to almost 25 weeks of full-time work at 40 hours per week. To put this in perspective, less than 1% of players on Lichess, a popular online chess platform, have played at least 5,000 games. Therefore, existing approaches for modeling individual behavior fall short of being full solutions to the problem, because they don't work for the vast majority of people. In order for everyone to benefit from algorithmically-informed teaching, learning, and collaboration, we first need another way to capture individual decision-making style—one that works in a much more data-efficient manner.

How could we go about modeling individual-level decision-making behavior for the everyday person with much more modest amounts of data available? This is a difficult task for two reasons. First, existing models for modeling human decision-making, such as Maia and Maia-2 (McIlroy-Young et al., 2022; Tang et al., 2024), are population models. This, as we will see, makes direct fine-tuning ineffective, especially for low-resource players. Second, human action prediction is formulated as a generative task to predict the next move that requires a model with strong generalization capabilities, which is particularly hard to achieve in a low-resource environment.

In this work, we propose a novel approach that produces **Maia4All**, a model that overcomes both of these challenges. Strikingly, **Maia4All** can model successfully individual-level play with only 20 games of data, in stark contrast to the previous requirement of at least 5,000 games. While Maia-2 shows virtually no progress when given 20 games of data played by a specific individual, and Maia fine-tuned with 1,000 games even gets worse than a base model, **Maia4All** significantly rises in accuracy from a baseline of 51.4% to 53.2%—a comparable rise to the accuracy gains reported in previous work using 5,000 games per player (McIlroy-Young et al., 2022). By this measure, our approach is 250 times more data-efficient than prior methods.

We achieve data-efficient modeling of individual behavior in chess with two methodological contributions. First, after showing that straightforward fine-tuning fails, we design a novel two-stage fine-tuning approach. First, we *enrich* Maia-2 by fine-tuning it to a diverse set of *prototype* players with rich game histories in order to adapt the model parameters from population-level modeling to individual-level modeling. Empirically, this makes it easier for the model to further adapt to low-resource players. In the second stage, we further fine-tune this prototype-infused model with low-resource player data. This two-stage approach is surprisingly effective—fine-tuning directly on low-resource players doesn't work, but first enriching the model with a carefully-selected set of prototype player data is key to our eventual success. Our second contribution is to start with a discriminative task instead of attempting the difficult generative task directly; we first find the most similar prototype player to the target player we want to model with a prototype-matching meta-network. Once we've identified a suitable prototype player, we initialize the target player's embedding with the prototype's embedding, and fine-tune on their limited data with this much better initialization.

Our framework provides state-of-the-art individual behavior modeling in chess, which can open the door to personalized chess teaching and learning. Beyond chess, our novel two-stage design has potential to

benefit other domains that require data-efficient adaptation to individual behavior. To demonstrate its broader applicability, we include a case study on idiosyncratic LLMs, showing how our two-stage optimization framework enables LLMs to mimic individual writers effectively from limited data.

## 2 Methodology

Our methodology consists of two major steps. Starting from a base model (Maia-2 in our case), we first conduct an *enrichment* step that enables the model to better capture individual-level patterns. Second, we perform a *democratization* step that adapts the model to work on unseen players with limited data. We present preliminary details about the base model followed by our two-step methodology.

### 2.1 Base Model: Maia-2

**Choosing a Base Model.** Maia-2 (Tang et al., 2024) is a state-of-the-art human-like chess AI designed to predict human moves across skill levels using a unified transformer architecture. Unlike traditional chess AIs such as AlphaGo (Silver et al., 2016) and AlphaZero (Silver et al., 2017), which focus on best-move prediction and optimal play, Maia-2 is explicitly trained to model human decision-making, making it a strong foundation for our work. Maia-2 improves upon its predecessor, Maia (McIlroy-Young et al., 2020), by learning a unified parameter space where move prediction is modulated through skill-level embeddings, rather than relying on separate models for different skill levels. This design makes Maia-2 an ideal base model for fine-tuning toward individual behavior modeling, as its skill-level embeddings, initially trained on population-level data, can be further adapted to capture the unique patterns that characterize individual-level play. Allie (Zhang et al., 2024) is another approach to human move-matching which introduced an adaptive Monte-Carlo Tree Search with pondering time prediction method. We chose Maia-2 as our base model for two reasons specific to individual modeling. First, Allie models skill levels with linear interpolation between a weak and a strong control token, which may be inadequate for capturing playing style that is beyond strength or non-linear in nature. Unlike Maia-2's skill-level embeddings that can be fine-tuned from population-level representations to adapt to individuals, Allie's interpolation mechanism does not provide a straightforward path for individual adaptation. Second, Allie requires a complete move history from the starting position for prediction, whereas Maia-2 operates on single board positions, which is more widely applicable in cases such as modeling individual puzzle solving, where move history from the starting position is often unavailable.

**Architecture.** We first present a concise overview of Maia-2's architecture. As shown in Figure 1, Maia-2 consists of a chess position encoder, a skill encoder, and transformer-based skill-aware blocks. The model takes a chess position $p$, represented as a multi-channel tensor, along with skill levels of the active player $r(a)$ and the opponent player $r(o)$ as inputs, where $r(\cdot)$ denotes the mapping from a player to its skill level. Following the original design of Maia-2, when modeling only the behavior of the active player without considering variations in opponent strength, we set $r(a) = r(o)$ to ensure a consistent skill representation. We denote the policy head prediction of Maia-2 for player $i$ as:

$$a = f(p, r(i)|\theta), \tag{1}$$

where $\theta$ represents the trainable parameters in Maia-2. A ResNet-based (He et al., 2016) position encoder first encodes chess positions into position embeddings. The position embeddings then undergo channel-wise patching and linear transformation (Tang et al., 2024), and are fed into the residual flows of transformer blocks. Within these blocks, Maia-2 performs skill-aware attention to fuse skill embeddings with the position encodings. Maia-2 applies a vanilla ViT feed-forward network and a Add & Norm component to obtain the output of each block and add it back to the model's residual stream. The model's output includes a policy head for move prediction, a value head for game outcome prediction, and an auxiliary information head for accelerating game rule learning from objective training goals. We focus exclusively on the policy head, as it is responsible for modeling individual player behavior, whereas the value and auxiliary heads in Maia-2 are mainly used as proxy rewards to guide the model.

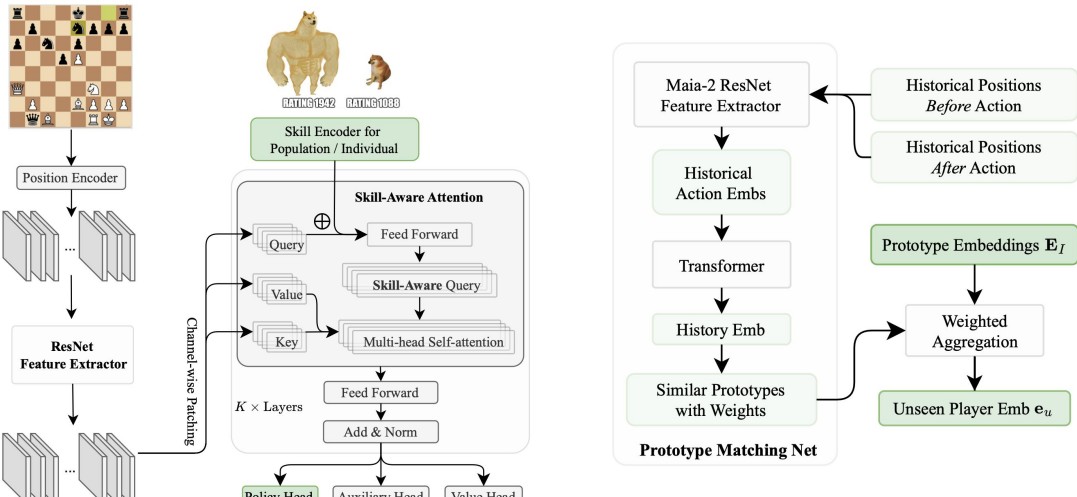

Figure 1: (Left) The architecture of our base model Maia-2, which uses population embeddings to adapt to different skill levels. We enrich Maia-2 by extending population embeddings to individual embeddings. (Right) The procedure of Prototype-Informed Initialization and the architecture of the Prototype Matching Network.

## 2.2 Enrichment Step

The first step of our methodology is an enrichment step that adapts the base model so it can capture individual-level behavior instead of only aggregated population-level patterns.

**Population Modeling in Maia-2.** Chess players can be meaningfully grouped by skill level (McIlroy-Young et al., 2020; 2022), which is quantified using widely adopted rating systems (Elo, 1967; 1978). Maia-2's skill encoder follows this norm and is designed for player populations at different skill levels. Let $\mathbf{E}_P \in \mathbb{R}^{|\mathbf{E}_P| \times d}$ be the matrix of population embeddings, where each row corresponds to the embedding with dimension $d$ of a group of players that share a similar strength:

$$\mathbf{E}_P = [\mathbf{e}_{(0,1100]}, \mathbf{e}_{(1100,1200]}, ..., \mathbf{e}_{(2000,+\infty)}]^\top. \tag{2}$$

Given a player $i$ of skill level $r(i)$, we look up the embedding matrix $\mathbf{E}_P$ by rows to map the player skill level to its embedding $\mathbf{e}_i = \mathbf{E}_P[r(i)]$. We decompose pre-trained Maia-2 parameters $\theta = \{\phi, \mathbf{E}_P\}$ for clarity:

$$a = f(p, r(i)|\phi, \mathbf{E}_P), \tag{3}$$

where $\phi$ denotes the model parameters except for the population embedding matrix. Given the universal set of parameters $\phi$, Maia-2 adapts to different player populations by dynamically selecting the corresponding population embedding, allowing the model to account for variations in player skill levels.

**Enriching Maia-2.** To extend Maia-2 beyond population-level modeling, we generalize population embeddings (e.g. one entry per entire rating range of players) to individual embeddings (e.g. one entry per player), enabling the model to capture variations in individual player behavior much more precisely. Given a set of individual players $I$ (the choice of which is still to be determined), we denote their embedding matrix as $\mathbf{E}_I \in \mathbb{R}^{|\mathbf{E}_I| \times d}$, where each row corresponds to a specific player. Given a player $i \in I$, we look up $\mathbf{E}_I$ by rows to obtain the corresponding embedding, which is initialized with the player's population embedding before the enrichment step: $\mathbf{e}_i = \mathbf{E}_I[i] \leftarrow \mathbf{E}_P[r(i)]$. The enriched Maia-2 model retains the same universal parameters $\phi$ as Maia-2 at initialization:

$$a = f(p, i|\phi, \mathbf{E}_I). \tag{4}$$

Since $\phi$ has already been trained to model diverse groups of players, it serves as a strong foundation for fine-tuning toward capturing individual-level variation. To enrich the model so it captures individual-level

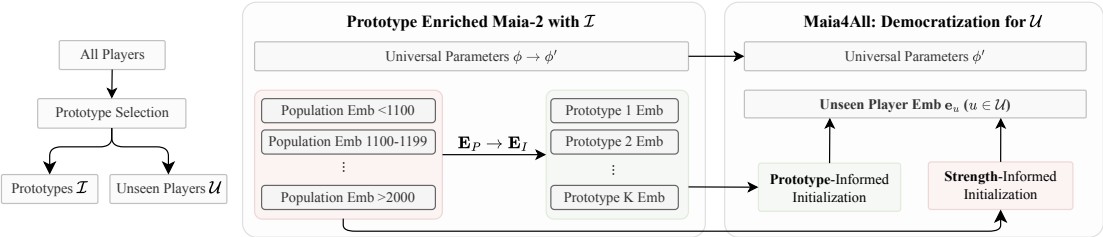

Figure 2: Overview of our proposed framework.

behavior, we minimize the following training objective:

$$\mathcal{L}(\phi, \mathbf{E}_I) = -\frac{1}{\sum_{i \in I} |\mathcal{P}_i|} \sum_{i \in I} \sum_{p \in \mathcal{P}_i} \log P(a^* | f(p, i | \phi, \mathbf{E}_I)), \tag{5}$$

where $\mathcal{P}_i$ denotes the set of historical positions of player $i$ and $a^*$ is the ground-truth move made by the player. After fine-tuning, the model refines both $\phi$ and $\mathbf{E}_I$, producing updated parameters $\phi'$ and $\mathbf{E}_I'$ in the Enriched Maia-2 model:

$$a = f(p, i | \phi', \mathbf{E}_I') \tag{6}$$

**Prototype Selection.** While this enriched model can be directly used to model the behaviors of a pre-defined set of players $I$, it is unrealistic to include all players (e.g., the 11 million players on Lichess). Firstly, learning separate embeddings for massive sets of players is challenging for computational and model learnability reasons. But even more importantly, most players haven't played enough games to support an accurately learned embedding for them. Furthermore, there are always new chess players, who would require constant re-training to support. Therefore, this enrichment step produces a crucial *intermediate* but incomplete model on the way towards our goal of truly efficient and effective individual behavior modeling. It will help facilitate further adaptation to all individual players, particularly those with sparse data.

Which individual players should be in our set $I$? This extensibility requirement guided us to carefully select players for this enrichment step with two key criteria. First, players should have sufficient historical data (i.e. completed games) to ensure that their decision-making styles can be well learned in $\mathbf{E}_I$, and so $\phi'$ will not be corrupted by under-trained embeddings. Second, the player set should be balanced across different skill levels to prevent the model from being biased toward any particular skill level. Ensuring a diverse distribution of players helps this enriched model learn a well-calibrated set of parameters $\phi'$ that generalizes effectively across individual players with varying levels of expertise. We refer to these carefully selected players as *prototypes*, and the enriched model trained on these prototypes as *Prototype-Enriched Maia-2*, which serves as a foundation for further adaptation to a much broader range of individual players.

## 2.3 Democratization Step

The enrichment step transitions model parameters $\phi'$ from being optimized for population-level modeling to being optimized for individual-level modeling. As previously discussed, it is infeasible to use the enriched model directly on all players, but it is now much more adaptable than the base model to *unseen* players $\mathcal{U}$—any one of the vast majority of players who were not selected as prototypes. The key reason for this adaptability is that $\phi'$ becomes more responsive to individual embeddings, enabling more precise modulation of move predictions. This effect arises because the skill embeddings, initially constrained to 11 population embeddings in the base model Maia-2, have been expanded to a much larger set of individual embeddings spanning a diverse player distribution, requiring the model to distinguish players with greater sensitivity.

We use Prototype-Enriched Maia-2 as an intermediate model to efficiently adapt to unseen individual players— even those with minimal data to learn from. This second and final step in our methodology *democratizes* Maia-2 and Prototype-Enriched Maia-2, making individual behavior modeling accessible to all players, not just those with extensive historical data. We call this model **Maia4All**.

To adapt to an unseen player $u$ from the intermediate model, we minimize the loss function:

$$\mathcal{L}(\mathbf{e}_u) = -\frac{1}{|\mathcal{P}_u|} \sum_{p \in \mathcal{P}_u} \log P(a^*|f(p|\phi', \mathbf{e}_u)), \tag{7}$$

where $\mathcal{P}_u$ denotes the set of historical positions of unseen player $u$, $a^*$ is the ground-truth move made by $u$, and $\mathbf{e}_u$ denotes the individual embedding for $u$. Thus, $\mathbf{e}_u$ is optimized towards modeling $u$, denoted as $\mathbf{e}'_u$, and the policy head prediction is given by:

$$a = f(p|\phi', \mathbf{e}'_u). \tag{8}$$

**Strength-Informed Initialization.** Although $\phi'$ lays a strong foundation for democratizing Maia-2 to unseen players, an effective initialization strategy for unseen player embeddings is crucial for data-efficient adaptation. To address this, we propose to initialize unseen player embeddings using prior knowledge, enabling more efficient parameter updates. A natural source of prior knowledge is player strength, as reflected by their ratings. In chess, decision-making style is strongly influenced by a player's strength. For example, a novice is unlikely to employ the deep strategic insights characteristic of more advanced players. Therefore, player strength serves as a natural reference point for initializing player embeddings:

$$\mathbf{e}_u \xleftarrow{\text{Initialize}} \mathbf{E}_P[r(u)]. \tag{9}$$

However, player strength is only one dimension of a richer space of behavior. People not only vary in the overall quality of their play, but also in their *decision-making style*. In chess, people can tend towards being more aggressive or defensive, positional or tactical, intuitive or calculating, etc. Although overall quality metrics like ratings capture a lot of variation, it is likely that decision-making style is more variable and requires more careful initialization. Therefore, we aim to initialize the unseen player embedding $\mathbf{e}_u$ with similar player embeddings.

**Prototype-Informed Initialization.** Beyond the more responsive universal parameters inherited from the intermediate model, the learned prototype embeddings $\mathbf{E}_I$ also play a crucial role in prior-informed initialization by leveraging similarities with existing prototypes. Since our prototype selection ensures a balanced distribution across skill levels, the prototype set $\mathcal{I}$ should ideally cover the diverse player styles across skill levels. Furthermore, because prototypes are chosen from players with extensive game histories, their embeddings are will be well-trained and serve as reliable references for initializing new player embeddings.

We train a transformer-based meta-network for prototype matching, i.e., given an unseen player, identifying similar prototypes that will be useful for initialization. As shown in Figure 1 (Right), during training, given a collection of historical moves from a prototype player $i \in \mathcal{I}$, we extract position features before and after the player's ground-truth actions using ResNet-based towers pre-trained by Maia-2. To encode player style, we apply stacked Transformer layers with mean pooling to aggregate historical action embeddings into a single history embedding. This embedding is then fed into a classification head with $|\mathcal{I}|$ outputs, designed to identify the corresponding prototype player. Since we use prototype players for training, the true identity of the player generating the history is naturally available, allowing us to train the $|\mathcal{I}|$-class classifier using a cross-entropy loss. We refer to this model as the **Prototype Matching Network** (PMN).

During inference, we input the historical moves of an unseen player $u \in \mathcal{U}$ into PMN to assess the similarity between the player's style and the prototype players. The model's predictions are passed through a softmax function, and the top-$k$ most similar prototypes are selected. Their embeddings are then combined using a weighted average to serve as the prototype-informed initialization of $\mathbf{e}_u$.

Note that both prototype matching and our final task human move prediction exploit the same set of historical behaviors of unseen players. However, prototype matching is a much easier task than the latter. This is because prototype matching is essentially a *discriminative* task against a fixed set of classes (prototypes), and human move prediction is a next-move *generative* task that requires a deeper understanding of the player's decision-making style. Therefore, we initialize the player embedding with the easier prototype matching task to get a rough understanding of how similar players behave and further calibrate the player embedding with human move prediction loss.

# 3 Experiments

We now conduct extensive experiments to evaluate how our novel two-step methodology for efficient individual-level behavior modeling performs and compares with strong baseline methods.

## 3.1 Experimental Settings

**Datasets.** Online chess platforms feature a variety of game types, including blitz, rapid, and classical, each representing games played at different time controls (amount of time given to each player for the whole game). We use data derived from the open database provided by Lichess, a well-known large open-source chess platform. In Lichess, since each game type is given a separate rating, ratings across different game types are not comparable (e.g. a rating of 1800 in "Rapid" is significantly weaker than a rating of 1800 in "Blitz" on Lichess). We focus on Blitz games because it is data-rich, and do not mix with other game types to ensure the ratings are meaningfully comparable. For prototype players, we use their game history in 2023 to compromise between the changing player strengths and styles over time and the availability of rich historical data. We follow Maia-2 by dividing players into 11 bins: under 1100, over 2000, and nine 100-point wide strength bins from 1100 to 2000, i.e., $|\mathbf{E}_P| = 11$. During training for Prototype-Enriched Maia-2, we use the game history of the $N$ most frequent players in each strength level, i.e., $|\mathbf{E}_I| = 11N$. For testing, we use 10 prototype players and 10 unseen players per strength level. To simulate unseen players with limited game history, we restrict their training positions to the first $M$ positions while evaluating their performance on the last 2048 positions recorded in 2023. This results in test datasets containing 225,280 positions for both prototype and unseen players.

**Implementation Details.** To maintain a consistent perspective from both sides of players, we used board flipping during both training and testing; that is, positions with black to move were mirrored so that all analyses could be conducted from the white side's viewpoint. Consistent with prior work, we further refined our dataset through game and position filtering, selecting games with available clock information and disregarding the initial 10 plies of each game as well as positions where either player had less than 30 seconds remaining. This filtering procedure mitigates the noise introduced by decisions made under extreme time pressure, which could skew the true representation of a player's strength and style. We report all hyperparameters involved in training in Table 7 in the Appendix.

**Evaluation Protocol.** We evaluate our method with top-1 move-matching accuracy, which is essentially an extensive human study: we observe what humans would play in natural situations recorded by the Lichess Database, and see if it matches the predicted move of our system. We also measure the perplexity of move predictions, which reflects the model's confidence in its predictions. A lower perplexity indicates the model is more confident and accurate in human move prediction, as it corresponds to a higher likelihood (lower log-likelihood) of the correct human move. We report the results with three categories: *Skilled* (Blitz rating up to 1600, which slightly exceeds the initial rating of 1500), *Advanced* (Blitz rating between 1600 and 2000), and *Master* (Blitz rating over 2000, roughly comprising the top 10% of players (Duplessis)).

**Baselines.** Maia (McIlroy-Young et al., 2020) is a set of 9 separate models, each trained on a different set of players at different skill levels from 1100 to 1900. Maia-1100 models the weaker players, Maia-1500 the intermediate players, and Maia-1900 the higher-skill players. We choose one of the Maia models for each population such that it is the nearest to their strength level for fair comparison. Since Maia-Individual (McIlroy-Young et al., 2022) is designed for data-rich settings, the published results of Maia-Individual indicate that it requires 5,000 games per player to show improvement over Maia. However, Maia4All, as a method specifically designed for low-resource individual behavior modeling, at most has access to 100,000 positions ($\approx$ 2,500 games). Therefore, we do not include Maia-Individual as a baseline, since it does not apply to the sparser settings we consider (and which cover the vast majority of players on Lichess). Note that the goal of meta learning methods (Finn et al., 2017; Nichol et al., 2018) is to optimize for the query set without emphasizing the performance on the support set (similar to prototype players). While in Maia4All, we consider all players equally, the separation of prototype and unseen players is for the purpose of democratizing the individual behavior modeling process for all players. Therefore, our work is not strictly under a meta-learning setting.

Table 1: Performance on unseen players under low-resource settings.

| | Move Prediction Accuracy ↑ | | | | Move Prediction Perplexity ↓ | | | |
|---|---|---|---|---|---|---|---|---|
| #Positions | 20,000 | 8000 | 2000 | 800 | 20,000 | 8000 | 2000 | 800 |
| #Games | ≈500 | ≈200 | ≈50 | ≈20 | ≈500 | ≈200 | ≈50 | ≈20 |
| Maia | 0.5132 | 0.5132 | 0.5132 | 0.5132 | 5.4530 | 5.4530 | 5.4530 | 5.4530 |
| Maia-2 | 0.5146 | 0.5146 | 0.5146 | 0.5146 | 4.5316 | 4.5316 | 4.5316 | 4.5316 |
| Maia-2-Strength | 0.5195 | 0.5196 | 0.5193 | 0.5189 | 4.4932 | 4.4939 | 4.4976 | 4.5022 |
| Maia4All-Strength | 0.5308 | 0.5298 | 0.5279 | 0.5249 | 4.3077 | 4.3238 | 4.3658 | 4.4151 |
| Maia4All-Prototype | **0.5365** | **0.5348** | **0.5334** | **0.5322** | **4.2295** | **4.2431** | **4.2669** | **4.2988** |

Furthermore, meta learning methods require split support and query sets (prototype and unseen players) beforehand. However, the process to separate prototypes and unseen players (prototype selection) is an inherent methodological component in Maia4All. Therefore, it's challenging to isolate Maia4All completely to apply classic meta learning methods as baselines directly.

## 3.2 Results

**Base model: Maia2.** The key difference between Maia and Maia-2 lies in their approach to modeling player's skill levels. Maia requires the selection of the appropriate sub-model based on a player's strength. In contrast, Maia-2 employs a unified modeling approach, allowing it to dynamically adapt to various skill levels within a single model. This adaptability is enabled by skill-aware attention, which modulates model predictions based on the corresponding skill embedding, which represents a population or an individual. For fair comparison, we select the sub-Maia model that matches each player's strength level. For Maia-2, skill embeddings are initialized using the corresponding population embedding that aligns with the player's strength level. As shown in Table 1 and Table 2, Maia-2 consistently outperforms Maia across all evaluation metrics and settings. While top-1 accuracy gains are important, they may overshadow more significant improvements in prediction quality. Such results show that Maia-2 can not only more accurately predict human behaviors but also be much more certain about its predictions. These results support our choice of adopting Maia-2 as the base model.

**Maia-2-Strength.** We developed a version of Maia4All that skips the enrichment step. Since the prototype embeddings are obtained within this step, prototype-informed initialization is not available. Therefore, we apply strength-informed initialization for fair comparison. We refer to this model as Maia-2-Strength. As shown in Table 1 and Table 2, directly fine-tuning from the base model for population modeling barely improves human move prediction accuracy and perplexity under low-resource and relatively data-rich settings; low-resource move matching only improves by 0.5 percentage points (p.p.) over the base model. This demonstrates the need for our two-stage approach and prototype-informed initialization.

**Prototype-Enriched Maia-2.** Previous work (McIlroy-Young et al., 2022) has shown that fine-tuning population models to individual players requires a substantial amount of data. Specifically, the previous state-of-the-art sees performance improvements over the base population model emerge only after fine-tuning on 5,000–10,000 games ($\approx$ 200,000–400,000 positions) per player, whereas with only 1,000 games ($\approx$ 40,000 positions), the fine-tuned model underperformed compared to the base model. In contrast, Prototype-Enriched Maia-2 significantly outperforms both Maia and Maia-2 across all evaluation metrics, where move prediction accuracy and perplexity are shown in Figure 3. Notably, our model achieves these improvements using a maximum of 100,000 positions ($\approx$ 2,500 games) per player—an amount that lies between the thresholds where Maia fine-tuning is ineffective (1,000 games) and where it becomes beneficial (5,000 games). These results highlight the effectiveness of our enrichment step, demonstrating that Prototype-Enriched Maia-2 requires far less historical data to accurately model individual decision-making styles while achieving superior performance. Furthermore, the strong results indicate that the universal parameters $\phi'$ successfully transition from population-level modeling to individual-level modeling, making them well-suited for the downstream

Table 2: Performance on unseen players with relatively rich histories (100,000 positions ≈ 2,500 games).

| Skill Categories | Move Prediction Accuracy ↑ | | | | Move Prediction Perplexity ↓ | | | |
| --- | --- | --- | --- | --- | --- | --- | --- | --- |
| | Skilled | Advanced | Master | Overall | Skilled | Advanced | Master | Overall |
| Maia | 0.4996 | 0.5099 | 0.5285 | 0.5132 | 5.8687 | 5.4642 | 5.1300 | 5.4530 |
| Maia-2 | 0.5008 | 0.5158 | 0.5364 | 0.5146 | 4.7900 | 4.4389 | 4.1936 | 4.5316 |
| Maia-2-Strength | 0.5071 | 0.5212 | 0.5400 | 0.5199 | 4.7264 | 4.4113 | 4.1760 | 4.4903 |
| Maia4All-Strength | 0.5226 | 0.5376 | 0.5478 | 0.5336 | 4.4504 | 4.1733 | 4.0291 | 4.2599 |
| Maia4All-Prototype | **0.5261** | **0.5408** | **0.5554** | **0.5381** | **4.4018** | **4.1048** | **3.9219** | **4.1899** |

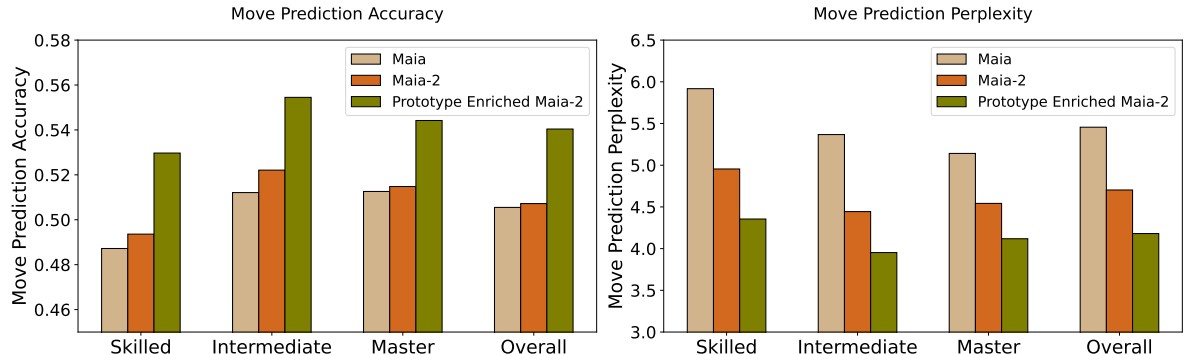

Figure 3: Move prediction accuracy (Left) and perplexity (Right) on prototype players.

democratization step to model unseen players. Additionally, these results support that the prototype embeddings are well-learned, ensuring they serve as effective references for prototype-informed initialization.

**Maia4All.** We limit the democratization step to access 100,000 historical moves to demonstrate the reduced size of historical data needed for achieving sufficient improvement. As shown in Table 2, the best performing Maia4All variant, Maia4All-Prototype, outperforms Maia with over 2.5 percentage points in accuracy and around 1.2 in perplexity (whereas Maia-Individual barely shows any improvement at this amount of data). These results demonstrate Maia4All's capability to adapt to unseen players with relatively rich data, and the amount of data needed is significantly lower. In the human move prediction problem for amateur players, the ceiling accuracy is far below 100% given the randomness and diversity of their decisions—even the same player won't always make the same decision when faced with the same position. Given this unpredictability, a 2.5 percentage point gain is a significant improvement—around half of the gain in amateur human move matching between Stockfish, the world's strongest chess engine (which obviously plays much differently than amateur humans) and base Maia, a model specifically designed to play like humans.

When even fewer historical behaviors are accessible, Maia4All can still adapt to unseen players with considerable improvement in move prediction accuracy and perplexity. In particular, with only 800 positions (20 games, which is considered incredibly few for human behavior modeling in chess), Maia4All can transfer its predictions to unseen players with over 1.9 more percentage points and 1.1 lowered perplexity with prototype-informed initialization. Note that the number of accessible positions is at most 20,000 positions (worth 500 games), whereas fine-tuning Maia with 1,000 games actually results in *negative* improvement (McIlroy-Young et al., 2022).We also show that Maia4All remains effective under incremental learning in extremely low-resource settings, where game history is added gradually, starting from no history. Also, Maia4All data efficiency does not come at the cost of performance with rich data. More details are presented in Appendix A.

**Prototype-Informed Initialization.** Strength-Init and Prototype-Init in Table 3 represent strength and prototype-informed initialization, respectively, without further adaptation in the democratization step. Prototype-Init consistently outperforms Strength-Init across all data settings, explaining the superior per-

Table 3: Performance of prior-informed initialization.

| #Positions | Move Prediction Accuracy ↑ | | | | Move Prediction Perplexity ↓ | | | |
|---|---|---|---|---|---|---|---|---|
| | 20,000 | 8,000 | 2,000 | 800 | 20,000 | 8,000 | 2,000 | 800 |
| Strength-Init | 0.5008 | 0.5008 | 0.5008 | 0.5008 | 4.8344 | 4.8344 | 4.8344 | 4.8344 |
| Prototype-Init | 0.5180 | 0.5175 | 0.5173 | 0.5167 | 4.5360 | 4.5333 | 4.5400 | 4.5459 |

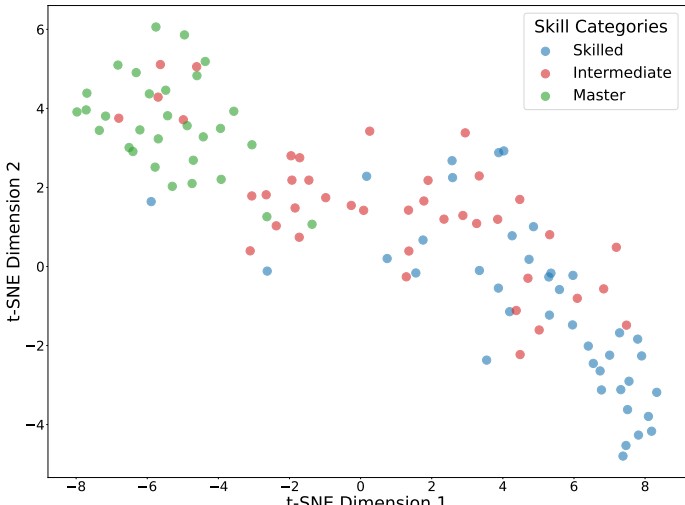

Figure 4: Visualization of prototype-informed initialized unseen player embeddings.

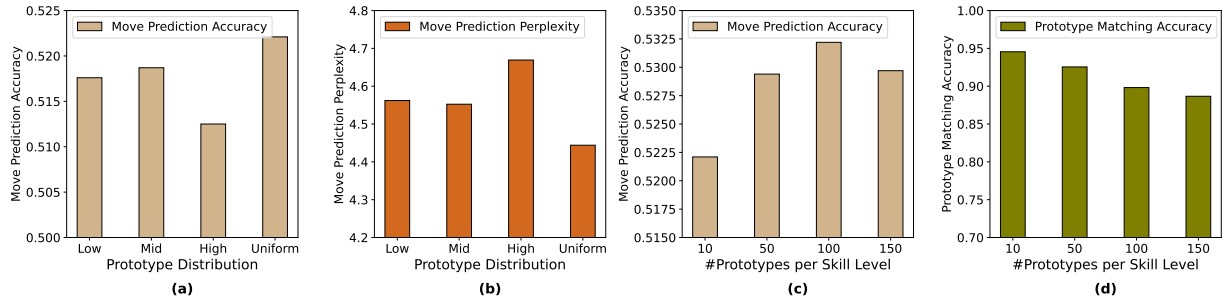

Figure 5: Effects of the distribution of prototypes (a,b) and prototype quantity per strength level (c,d).

formance of Maia4All-Prototype over Maia4All-Strength. These results highlight the effectiveness of the Prototype Matching Network and reinforce the importance of better initialization—demonstrating that a strong foundation learned from a discriminative task (prototype matching) can enhance the final performance of the generative task (next-move prediction).

We take a step further to examine the patterns of prototype-informed initialized unseen player embeddings using t-SNE for dimensionality reduction. As shown in Figure 4, the embeddings exhibit a linear structure in 2D space, with Skilled and Master players forming clusters at opposite ends and Intermediate players concentrated along the middle. This indicates that prototype-informed initialization effectively encodes player strength. Additionally, the dispersion of embeddings around cluster centers suggests that the prototype-informed initialization also captures variations in individual playing styles.

**Hyperparameter Study.** The distribution of the prototypes to be matched is a hyperparameter. As shown in Figure 5 (a) and (b), if we only include the prototypes from a biased distribution of the population, i.e,

only select from low/medium/high-level players, it will result in lowered move prediction accuracy and raised perplexity compared to uniformly select $N$ prototypes from each strength level. Such results support our design choices of selecting the prototypes uniformly to cover the population space.

The number of prototypes $N$ per strength level is also a hyperparameter. Choosing an appropriate $N$ needs to compromise between the representativeness of prototypes for each range, i.e., more prototypes can better cover the player embedding space, and the difficulty in prototype matching, i.e., more prototypes means more candidates to be classified against. This is evidenced by the results shown in Figure 5 (c) and (d). We evaluate the top 1 matching accuracy of prototypical players under low-resource settings (800 positions). Increasing $N$ from 10 to 150 yields gradually lowered performance in prototype matching, while the best-performing Maia4All is achieved with a trade-off between prototype matching accuracy and player embedding space coverage.

## 4 Related Work

**Human Behavior Modeling in Chess.** The challenge of creating a chess engine that can outplay any human was solved over 20 years ago. The research focus then shifted towards the problem of extracting useful knowledge from these superhuman systems. A direct way of doing this is probing an AI chess engine in a human representation space. Evidence of human chess concepts learned by AlphaZero can be found and measured by linear probes (McGrath et al., 2022); furthermore, AlphaZero also encodes knowledge that extends beyond existing human knowledge but can be successfully taught to humans (Schut et al., 2023). Another direction is the creation of 'behavioral stylometry' models that can identify chess players from the moves they play (McIlroy-Young et al., 2021). Moreover, efforts have been made towards creating systems that strive for human-likeness over sheer strength (McIlroy-Young et al., 2020; Jacob et al., 2022; Tang et al., 2024; Zhang et al., 2024) in which models are trained to predict the next move a human will play, instead of optimizing for winning the game. In addition to predicting human actions at the population level, the models have been fine-tuned for individual-level human behavior modeling in *data-rich* settings (McIlroy-Young et al., 2022). Recent work (Omi et al., 2025) frames individual behavior modeling as multi-task learning, using parameter-efficient fine-tuning with shared LoRA adapters and per-player routing vectors; this approach focuses on generalization across domains (chess, Rocket League, and image generation) and style manipulation, rather than democratizing individual modeling to low-resource settings.

**Few-shot Learning and Meta Learning.** Few-shot learning focuses on the ability of models to learn and generalize from a very limited amount of labeled training data (Fei-Fei et al., 2006; Fink, 2004; Wang et al., 2020). Modeling unseen players follows the few-shot learning paradigm, where players' behavioral patterns are revealed by a limited collection of historical behaviors. Meta learning is a main approach to few-shot learning, aiming to improve novel tasks' performance by training on similar tasks. Meta learning can be categorized into metric-based methods (Vinyals et al., 2016; Snell et al., 2017; Koch et al., 2015; Sung et al., 2018) that aim to learn a similarity or distance function over objects and represent the relationship between inputs and the task space, model-based methods (Santoro et al., 2016; Munkhdalai & Yu, 2017), which focus on designing models with internal mechanisms to quickly adapt to new tasks, and optimization-based methods (Ravi & Larochelle, 2016; Finn et al., 2017; Nichol et al., 2018; Raghu et al., 2019) that aim to learn an initialization such that the model can adapt faster with few examples from there.

In Maia4All, our enrichment step follows multi-task pretraining methodologies common in optimization-based meta-learning, where training across diverse tasks (in our case, prototype players) produces universal parameters for rapid fine-tuning (Finn et al., 2017; Nichol et al., 2018; Wang et al., 2024). The democratization step combines two established techniques. First, context-only optimization: we freeze universal parameters and optimize only low-dimensional player embeddings, following the parameter-efficient adaptation paradigm demonstrated in meta-learning for context adaptation and domain transfer (Zintgraf et al., 2019; Houlsby et al., 2019). Second, prototype-based initialization: we use a learned prototype matching network to initialize embeddings based on behavioral similarity, following metric-based meta-learning approaches that leverage prototypes for few-shot adaptation (Snell et al., 2017; Triantafillou et al., 2019; Rusu et al., 2018; Sung et al., 2018).

Table 4: Performance of Maia4All-Prototype with optimized or frozen universal parameters $\phi'$.

|  | Accuracy ↑ | | Perplexity ↓ | |
| --- | --- | --- | --- | --- |
| #Positions | 20,000 | 800 | 20,000 | 800 |
| Optimized | 0.5395 | 0.5297 | 4.1949 | 4.3753 |
| Frozen | 0.5365 | 0.5322 | 4.2295 | 4.2988 |

While these techniques are established, figuring out how to successfully combine several techniques into a pipeline that works surprisingly well for individual behavior modeling in chess is challenging. Such a specific combination for modeling human behavior represents a novel contribution motivated by chess's two-dimensional behavioral structure (strength and style). We require enrichment because population-level embeddings cannot capture individual playing styles within rating bins; we require prototype-based initialization because stylistically similar players can exist across different skill levels, necessitating a learned matching mechanism beyond rating-based lookup; and we require context-only optimization to preserve universal chess knowledge while adapting to each player's coordinates in the behavioral space under low data. To our knowledge, this is the first work that integrates prototype enrichment, learned prototype matching, and context optimization in a unified framework for individual behavior modeling in strategic decision-making domains.

**Imitation Learning.** Our work can also be viewed as related to the imitation learning literature, where models are trained to perform tasks after observing expert human demonstrations Schaal (1999); Zare et al. (2024); Wang et al. (2019). In the imitation learning context, the model is usually attempting to learn a value function (inverse RL) (Ng et al., 2000), or to quickly learn an optimal solution to a given optimization problem (Schaal, 1999). In this paper, we attempt to learn a flawed, human value function using *non-expert* demonstrations. Additionally, many imitation learning methods require the model to be in the same, or similar, state to the demonstrated one (Ho & Ermon, 2016; Zare et al., 2024) which is a condition that is impossible to guarantee in chess outside of the early game.

## 5 Discussion

**Behavioral Stylometry.** The prototype matching network can be directly used for behavioral stylometry (McIlroy-Young et al., 2021), i.e., identifying players given their historical behaviors. Since we freeze the shared parameters $\phi'$ and only optimize player embeddings for unseen players, the player embeddings are directly comparable within the same embedding space. Therefore, our design supports behavioral stylometry off the shelf. As shown in Figure 5, with only 800 positions (around 20 games), our model can identify the player with 89% accuracy with 1 shot from 1100 candidates (100 players per strength level with 11 levels).

**Parameter Efficient Fine-Tuning.** During the democratization step, we freeze the universal parameters $\phi'$ and optimize only the unseen player embedding $\mathbf{e}_u$. Table 4 compares this approach with an alternative where $\phi'$ is also optimized, revealing two key insights. First, when the unseen player has a very limited game history, freezing $\phi'$ leads to better performance, suggesting that it acts as a form of normalization to prevent overfitting, which is particularly beneficial in extremely low-resource settings. Second, when more historical data is available—though still within a low-resource regime—optimizing $\phi'$ provides a marginal improvement over freezing it. However, fine-tuning only $\mathbf{e}_u$ remains significantly more parameter-efficient while achieving comparable results. This follows the principles of parameter-efficient tuning seen in the LLM literature, such as Prompt-Tuning (Lester et al., 2021) and Prefix-Tuning (Li & Liang, 2021). Given these findings, our original design of freezing $\phi'$ ensures better scalability for democratizing *large* models to individuals with minimal computational overhead while maintaining strong performance.

**Maia4All Generalization.** To demonstrate the broader capability of our approach beyond chess, we explore the generalization towards Large Language Model (LLM) adaptation to low-resource authors, which we term

Table 5: Performance comparison of prototype-enriched prompt tuning under different training settings for idiosyncratic LLM.

| #Tokens | LM Loss ↓ | | | Perplexity ↓ | | |
|---|---|---|---|---|---|---|
| | 1000 | 2000 | 3000 | 1000 | 2000 | 3000 |
| 1-step | 2.794 | 2.772 | 2.740 | 17.189 | 16.839 | 16.318 |
| 2-step | **2.792** | **2.757** | **2.724** | **17.131** | **16.546** | **15.996** |

*idiosyncratic LLM*, as a case study. The key components of our method find natural analogs in this domain: chess positions correspond to text sequences, player styles to writing styles, and next-move predictions to next-token predictions.

Following our two-stage approach, we first select 100 authors with the richest text content from Project Gutenberg (Project Gutenberg Literary Archive Foundation, 2003) as prototypes. We then enrich a base LLaMA-3.1-8B (Meta AI, 2024) model by extending its vocabulary with author-specific tokens and fine-tuning both the token embeddings and a set of LoRA parameters (Hu et al., 2021) dedicated to individualizing writing styles. This process is analogous to how we enrich Maia-2 with prototype player embeddings. Simultaneously, we train a ModernBERT-based (Warner et al., 2024) Prototype Matching Network (PMN) that learns to identify stylistic similarities between text samples. For new authors, we initialize their token embeddings through prototype matching and fine-tune only these embeddings while keeping other parameters fixed, similar to the democratization step. We select test authors who are *genuinely* low-resource for the base model. Starting from authors with the most limited texts in Project Gutenberg, we further identify the authors for whom the base model exhibits higher language modeling loss than our prototype-enriched model's average validation loss. This criterion ensures we focus on authors whose writing styles are sufficiently unfamiliar to the base model, resulting in a pool of 30 challenging cases.

Table 5 shows consistent improvements in language modeling loss and perplexity when tuning author embeddings with our prior-informed initialization (2-step) compared to direct tuning with base model (1-step), across different low-resource data settings from 1,000 to 3,000 tokens. These results suggest the effectiveness of our framework in capturing individual characteristics from limited data generalizes beyond its original domain in chess, demonstrating the potential in various domains requiring personalized behavior modeling. While we use language modeling loss as a proxy for style adaptation, future work could explore more direct metrics for evaluating writing style transfer, such as author-specific linguistic features or evaluation of stylistic similarity from human expert. Details of our implementation and evaluation for this case study of idiosyncratic LLM can be found in Appendix C.

## 6 Conclusion

We introduce Maia4All, a data-efficient framework for modeling individual behavior through an enrichment step and a democratization step. Maia4All can effectively capture individual playing styles even in extremely low-resource settings. The successful extension of our framework to modeling individual writing styles in LLMs suggests that our proposed strategy could potentially generalize beyond chess, offering a promising approach for personalizing AI systems in more domains.

## 7 Limitation

While our framework achieves significant improvements in data efficiency, it requires a set of prototype players with rich historical data for the enrichment step. This dependency might limit its applicability in newer domains where such extensive behavioral traces are not yet available. Additionally, although our LLM case study demonstrates potential generalizability, we primarily validate our approach in chess, where the action space is discrete and well-defined. While Maia4All represents meaningful progress in modeling human decision-making, future work should demonstrate whether these gains translate to practical benefits in downstream applications, such as personalized teaching and human-AI collaboration.

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

## A   Additional Experiments

| #Position | #Games | Accuracy | Perplexity |
|-----------|--------|----------|------------|
| 0         | 0      | 0.5146   | 4.5316     |
| 100       | 2.5    | 0.5252   | 4.4158     |
| 200       | 5      | 0.5271   | 4.3733     |
| 300       | 7.5    | 0.5285   | 4.3435     |
| 400       | 10     | 0.5292   | 4.3331     |
| 500       | 12.5   | 0.5302   | 4.3209     |
| 600       | 15     | 0.5309   | 4.3119     |
| 700       | 17.5   | 0.5314   | 4.3078     |

Table 6: Maia4all performance with gradually added history on the extremely low-resource setting.

**Incremental Learning under Low-Resource.** We conduct additional experiments to show how performance evolves with gradually added history. In particular, we focus on the extremely low-resource setting. Since Maia4All operates on positions instead of games, we gradually add 100 positions ( 2.5 games) each time. As the results shown in Table 6, with even very little game history, Maia4All could still adapt to individual players effectively with almost a full percentage point gain (51.46% —> 52.52%). Furthermore, the improvement over the base model monotonically increases. Therefore, it is feasible to do prototype matching with very few initial observations without strength initialization, especially with our design that focuses on data-efficient learning: using the same set of historical positions for the easier discrimination (prototype matching for initialization) first, and for the harder move generation (Maia-2 fine-tuning) afterward.

**Individual Behavior Modeling with Rich Data.** Data efficiency does not come at the cost of performance. We ran Maia4All with 5000 game histories and achieved 56.2%, an almost 5 percentage point gain from the base model performance of 51.4%. Note that fine-tuning Maia using 5000 games per player improved from 53.2% to 55.0% with a 1.8 percentage point gain. These comparisons are meaningful but have the caveat that there is a test set mismatch caused by the reproducibility of Maia-Individual (McIlroy-Young et al., 2022). But we can still observe that Maia4All improved from a worse starting baseline (51.4% vs 53.2%) to a much better resulting performance (56.2% vs 55.0%).

## B   Maia4All Reproductibility

**Position Representation and Encoding.** We follow the well-established prior works (McIlroy-Young et al., 2020; Silver et al., 2017) to represent chess positions as multi-channel $8 \times 8$ matrices, including:

- Piece Representation: The first 12 channels categorize the board's pieces by type and color, with one channel each for white and black Pawns, Knights, Bishops, Rooks, Queens, and Kings. A cell is marked 1 to denote the presence of a piece in the corresponding location, and 0 otherwise.

- Player's Turn: A single channel (the 13th) indicates the current player's turn, filled entirely with 1s for white and 0s for black, providing the model with context on whose move is being evaluated.

- Castling Rights: Four channels (14th to 17th) encode the castling rights for both players, with the entire channel set to 1 if the right is available or 0 otherwise.

- En Passant Target: The final channel (18th) marks the square available for en passant capture, if any, with 1 and 0s elsewhere.

One important departure from previous work is that we only use the current chess position, and not the last few chess positions that occurred in the game (models have typically incorporated the six most recent positions in the game). Many games with perfect information, including chess, can be modeled as alternating

Table 7: Hyperparameter Settings. We follow the notations from Maia-2 (Tang et al., 2024).

| | |
|---|---|
| Initial learning rate | $1e^{-4}$ |
| Weight decay | $1e^{-5}$ |
| Batch size (positions) | 8192 |
| Minimum move ply | 10 |
| Maximum move ply | 300 |
| Remaining seconds threshold | 30 |
| #Backbone blocks $K_{Conv}$ | 12 |
| #Attention block $K_{Att}$ | 2 |
| #Input channels $C_{\text{input}}$ | 18 |
| #Intermediate channels $C_{\text{mid}}$ | 256 |
| #Encoded channels $C_{\text{patch}}$ | 8 |
| Player embedding dimension $d$ | 128 |
| Attention head dimension $d_h$ | 64 |
| Attention intermediate dimension $d_{\text{att}}$ | 1024 |
| #Attention heads $h$ | 16 |
| player per range $N$ | 100 |

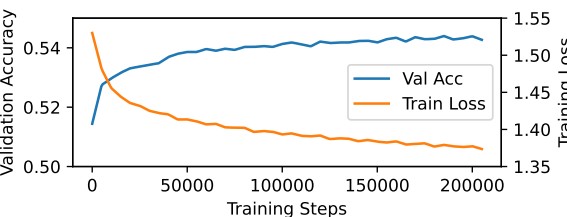

Figure 6: Training dynamics of the Prototype-Enriched Maia-2 model

Markov games (Littman, 1994; Silver et al., 2016), where future states are independent of past states given the current game state. Therefore, the current chess position theoretically encapsulates all the information necessary to make future decisions. Although human decision-making in chess may sometimes subtly depend on the historical lead-up to the current position, these effects are anecdotally small.

In exchange, we gain two large practical benefits. First, modeling AI-human move matching in a Markovian way vastly improves training *efficiency* by reducing the computational load via significantly smaller data usage for each decision. Second, it also enhances *flexibility*, enabling our resulting model to make predictions even without historical data, which is particularly advantageous in situations where only the current position is available, like chess training puzzles or any position that didn't necessarily occur in a full game.

**Enrichment Step Training Results.** We present detailed training dynamics of our Prototype-Enriched Maia-2 model in Figure 6. The training process exhibits smooth optimization trajectory suggests that our enrichment step successfully adapts the base model parameters to capture individual-level behavior patterns. The stable convergence is particularly noteworthy given that we are simultaneously training both the universal parameters $\phi$ and a large set of individual embeddings $\mathbf{E}_I$, demonstrating the effectiveness of our prototype selection criteria in ensuring model stability.

## C   Idiosyncratic LLM Reproducibility

**Dataset.** We use the Project Gutenberg dataset, which contains a vast collection of public domain books. Each book is preprocessed by cleaning line breaks and whitespace, then chunked into sequences of 2048 tokens with no overlap for efficient processing. We extract author information from the book metadata, filtering

Table 8: Hyperparameter Settings for Prototype-Enriched Training.

| | |
|---|---|
| Base Model | LLaMA 3.1 8B |
| LoRA Rank | 128 |
| LoRA Target Modules | Attention + FFN |
| LoRA Alpha | 32 |
| LoRA Dropout | 0 |
| Learning Rate | 1e-4 |
| Embedding Learning Rate | 5e-5 |
| Weight Decay | 0.01 |
| Batch Size | 16 |
| Gradient Accumulation Steps | 8 |
| Training Epochs | 2 |
| Warmup Ratio | 0.1 |
| Max Sequence Length | 2048 |

Table 9: Hyperparameter Settings for Idiosyncratic LLM.

| | |
|---|---|
| Initial learning rate (base) | $1e^{-4}$ |
| Weight decay | $1e^{-5}$ |
| Number of prototypes | 100 |
| Number of Training Tokens | 1K, 2K, 3K |
| Number of Testing Tokens | 5K |
| Top-k prototypes for matching | 2 |
| Temperature for prototype matching | 0.5 |
| Training Steps | 100 |

out anonymous works and those with ambiguous authorship. From all others in the dataset, we select 100 prototypes based on two criteria:

- having the largest amount of publications in Project Gutenberg to ensure prevalence

- having sufficient text content (at least 1000 chunks) to ensure reliable style learning

**Enrichment Step in LLM.** We implement our two-stage framework using LLaMA 3.1 8B as the base model. Our implementation focuses on efficient adaptation while preserving the model's core capabilities. In the first stage, we extend the model's vocabulary with author-specific tokens (e.g., `<author_Jack London>`), creating explicit anchor points for learning author-specific writing styles. We employ LoRA for parameter-efficient fine-tuning, with hyperparameters listed in Table 8. The LoRA adaptation strategically targets key attention modules and feed-forward layers, allowing the model to learn author-specific transformations while maintaining its general language understanding capabilities. The training dynamics shown in Figure 7 demonstrate steady convergence, which suggests effective learning of author-specific styles while maintaining the model's general language capabilities.

To enable the prototype matching mechanism analogous to our chess implementation, we train a ModernBERT-based Prototype Matching Network (PMN). The PMN learns to map text sequences to a space where stylistic similarities between authors can be effectively measured. By training on chunks of 2048 tokens from each prototype author's works, the PMN achieves 94.7% accuracy on prototype classification. This high accuracy, compared to the 1% random baseline with 100 prototype authors, demonstrates the network's strong capability in discriminating distinct writing styles.

**More Experimental Results.** As shown in Table 5, this two-stage approach consistently outperforms direct fine-tuning across different data settings. Across 1000 to 3000 tokens of training data, which can be

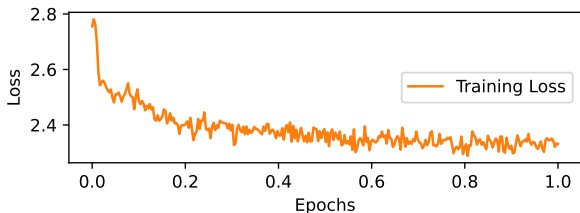

Figure 7: Training dynamics of the Prototype-Enriched Llama 3.1 8B.

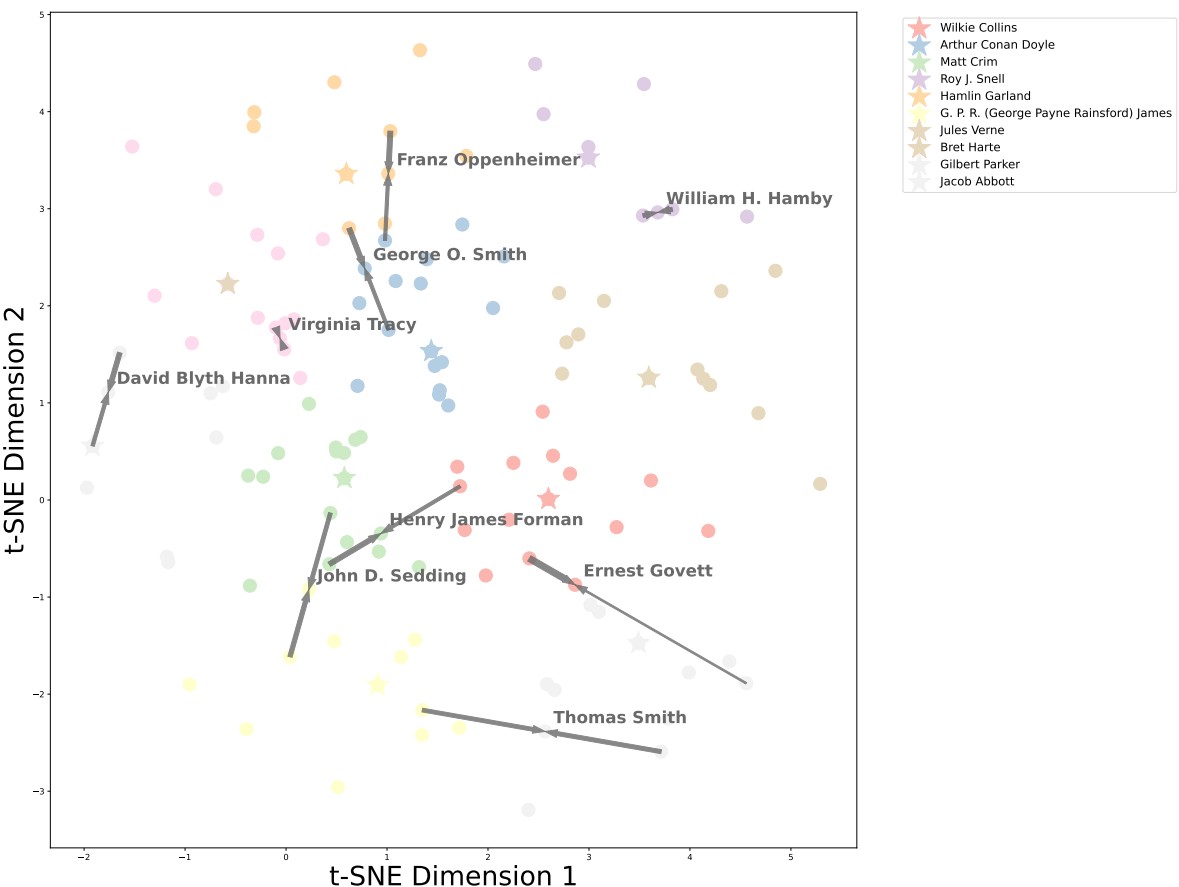

Figure 8: t-SNE visualization of author embeddings. Prototype authors are shown as dots, while new authors (labeled) are positioned based on their weighted combinations of prototype embeddings. Thicker arrows indicate stronger prototype influences.

fairly regarded as low-resource settings for LLM style transfer, our method shows improved language modeling performance, mirroring the benefits we observed in chess with increasing numbers of games. Furthermore, Table 10 shows the prototype matching results for new authors using 2000 training tokens. For each author, we list their top-2 matched prototypes and corresponding weights, alongside the language modeling loss with and without prototype initialization. The matches reveal interpretable stylistic connections that often align with historical and literary contexts. For instance, Herbert Strang and Richard Stead, who wrote adventure stories for young readers, is matched with Robert Louis Stevenson (57.2%) and Anthony Hope (42.8%). This pairing is particularly apt as Stevenson's adventure novels (like "Treasure Island") and Hope's romantic adventures (like "The Prisoner of Zenda") closely align with the literary style and target audience of Herbert Strang and Richard Stead. The corresponding improvement in language modeling loss (from 2.5114 to 2.4885) quantitatively validates this stylistic matching.

Table 10: Prototype Matching Results with 2000 Training Tokens.

| New Author | w/o Proto | w/ Proto | Matched Prototypes | Weights |
|---|---|---|---|---|
| George O. Smith | 2.7725 | 2.7581 | Richard Harding Davis, Jack London | 0.540, 0.460 |
| John D. Sedding | 2.7847 | 2.7652 | John Ruskin, Mrs. Humphry Ward | 0.618, 0.382 |
| Franz Oppenheimer | 2.6181 | 2.6307 | H. G. Wells, Upton Sinclair | 0.544, 0.456 |
| Virginia Tracy | 2.9959 | 2.9599 | Mrs. Humphry Ward, Edith Wharton | 0.622, 0.377 |
| Henry C. Merwin | 2.4606 | 2.4401 | William Dean Howells, Grant Allen | 0.509, 0.490 |
| Henry James Forman | 2.7958 | 2.7781 | Eugène Sue, Mór Jókai | 0.556, 0.444 |
| William H. Hamby | 2.7050 | 2.6828 | Emerson Hough, Allen Chapman | 0.601, 0.399 |
| A Pakeha Maori | 2.6287 | 2.6290 | Grant Allen, Hilaire Belloc | 0.674, 0.326 |
| Ernest Govett | 2.5229 | 2.5238 | John Ruskin, Richard F. Burton | 0.826, 0.174 |
| Vivia Hemphill | 3.0065 | 2.9972 | Bret Harte, Charles King | 0.546, 0.454 |
| Arthur Henry Chamberlain | 2.6130 | 2.6210 | Mór Jókai, Hilaire Belloc | 0.509, 0.491 |
| Samuel T. Pickard | 2.5931 | 2.5971 | Laura E. Richards, Oliver Optic | 0.689, 0.310 |
| Ruby K. Polkinghorne et al. | 2.5960 | 2.5691 | August Strindberg, Angela Brazil | 0.590, 0.410 |
| Thomas Smith | 2.6182 | 2.6006 | The Chautauquan LSC, Sir Walter Scott | 0.572, 0.428 |
| William Q. Judge | 2.7304 | 2.7389 | Upton Sinclair, Grant Allen | 0.534, 0.465 |
| Edith A. Browne | 2.5726 | 2.5413 | Grant Allen, Mór Jókai | 0.669, 0.331 |
| Donald Allen Wollheim | 2.6725 | 2.6612 | Jack London, Hamlin Garland | 0.502, 0.497 |
| Susan Carleton Jones | 2.7170 | 2.6796 | Rudyard Kipling, Edmund Yates | 0.645, 0.355 |
| Wallace Irwin | 3.8899 | 3.8506 | H. G. Wells, Carolyn Wells | 0.507, 0.493 |
| Hector MacQuarrie | 2.7740 | 2.7712 | William Le Queux, Hilaire Belloc | 0.567, 0.433 |
| S. Pérez Triana | 2.7608 | 2.7751 | Hilaire Belloc, Grant Allen | 0.554, 0.446 |
| François-Joseph Fétis | 2.4629 | 2.4748 | John Ruskin, William Dean Howells | 0.506, 0.493 |
| Herbert Strang and Richard Stead | 2.5114 | 2.4885 | Robert Louis Stevenson, Anthony Hope | 0.572, 0.428 |
| Matt Crim | 2.4610 | 2.4246 | Robert W. Chambers, Hamlin Garland | 0.617, 0.383 |
| William Chauncey Bartlett | 3.1245 | 3.0846 | Rudyard Kipling, Charles G. D. Roberts | 0.581, 0.418 |
| David Blyth Hanna | 2.9310 | 2.9009 | Anthony Hope, Mrs. Humphry Ward | 0.562, 0.438 |
| Edith Elise Cowper | 2.9251 | 2.9086 | Anthony Hope, L. T. Meade | 0.504, 0.496 |
| Ray Bradbury | 3.1877 | 3.1587 | Emerson Hough, August Strindberg | 0.665, 0.335 |
| Mary Elizabeth Hall | 2.5879 | 2.5707 | Margaret Vandercook, Grant Allen | 0.513, 0.487 |
| Dutton Payne | 3.1498 | 3.1341 | Laura E. Richards, James Grant | 0.615, 0.385 |

