# OpenReview forum: "Learning to Imitate with Less: Efficient Individual Behavior Modeling in Chess"
_TMLR — Accepted by TMLR_

### Review · Reviewer_UMFK · 2025-08-06

**Summary Of Contributions:**

This paper introduces Maia4All, a framework for customizing the behavior of Maia-2 (a humanlke chess model) to users with very few chess games (~20). The approach uses a two-stage optimization process: (1) an enrichment step that fine-tunes Maia-2 on selected "prototype" players to teach models to make use of individual "prototype" embeddings, and (2) a democratization step that specializes a model to a small number of unseen player games.

Strengths:
- Sample efficiency (individual behavior modeling in 20 games)
- Some evidence of generalization of the technique demonstrated through modeling author style with limited texts

Weakness:
- Modest improvements (51.4% -> 53.2%)
- Experiment setup is somewhat unrealistic (a practical system requires continual learning / test-time training), though should be easily addressable

**Audience:**

Yes

**Audience Explanation:**

Individual behavior modeling / continual learning is an important topic in machine learning. If the proposed technique generalizes, the paper should be valuable to a broad ML audience.

**Claims And Evidence:**

Yes

**Claims Explanation:**

**Sample Efficiency**

Prior work (Maia-Individual) required 5,000 games per player to show improvement over base models, which would take months of game play and is completely unrealistic. The enrichment training + prototype-matching initialization approach enables fast (~20 games) adaptation to user behavior, a huge improvement over what was previously possible.

An additional benefit of the method is that the model is not tuned during the "democratization" stage. So the amount of information stored is just a single embedding per user.

**Generalization**

Appendix B reports encouraging results on generalization of the proposed method to language modeling.
The proposed method reduces perplexity on low-resource authors after initializing from embeddings of similar authors.

**Limitations of Experimental Setup**

The evaluation assumes a batch learning scenario where all user games are available upfront for fine-tuning. Real-world deployment would likely require online/continual learning approach, where the model adapts incrementally as users play more games. While the strong performance with just 20 games suggests online adaptation should be feasible, the paper lacks analysis of:
- How performance evolves with sequential game-by-game updates
- Whether prototype matching remains reasonable effective with very few initial games (e.g., 1-5 games)

**Requested Changes:**

**Online learning experiment (minor)**

Real deployment scenarios would likely involve incremental learning as a previously unknown user plays more moves. Add discussion or preliminary analysis of:
- How performance evolves with move-by-move updates
- Whether it's feasible to do prototype matching with very few initial observations, or whether blending prototype / strength initialization is required (since there are no moves for prototype matching network to classify initially)

---

> ### Author Response · Authors · 2025-10-15
>
> We conduct additional experiments to show how performance evolves with gradually added history following your suggestion [**Limitation 1 and Requested Changes 1**]. In particular, we focus on the extremely low-resource setting that was suggested in the review [**Limitation 2**]. Since Maia4All operates on positions instead of games, we gradually add 100 positions (~2.5 games) each time. As the results shown in the following table, with even very little game history, Maia4All could still adapt to individual players effectively with almost a full percentage point gain (51.46% —> 52.52%). Furthermore, the improvement over the base model monotonically increases. Therefore, it’s feasible to do prototype matching with very few initial observations without strength initialization [**Requested Changes 2**], especially with our design that focuses on data-efficient learning: using the same set of historical positions for discrimination (prototype matching for initialization) first, and for generation (Maia-2 fine-tuning) afterward. We would like to add these results and the above discussion in the revision.
>
> | #Position | #Games | Accuracy | Perplexity |
> |----------|-------|----------|------------|
> | 0 | 0 | 0.5146 | 4.5316 |
> | 100 | 2.5 | 0.5252 | 4.4158 |
> | 200 | 5 | 0.5271 | 4.3733 |
> | 300 | 7.5 | 0.5285 | 4.3435 |
> | 400 | 10 | 0.5292 | 4.3331 |
> | 500 | 12.5 | 0.5302 | 4.3209 |
> | 600 | 15 | 0.5309 | 4.3119 |
> | 700 | 17.5 | 0.5314 | 4.3078 |

---

### Review · Reviewer_sV18 · 2025-08-22

**Summary Of Contributions:**

This paper introduces Maia4All, a  highly data-efficient framework for modeling individual human behavior in chess, addressing the critical limitation of previous methods that required prohibitively large amounts of data.
The main contributions can be summarized as follows:
- A Two-Stage Optimization Framework: The core contribution is the two-stage methodology consisting of an Enrichment Step and a Democratization Step. The enrichment step first creates a model conditioned on individual styles using data-rich "prototypes," and then leverages this model to rapidly personalize for new, low-data users.
- Experimental results show nice improvement on the data efficiency: The paper demonstrates that Maia4All can achieve significant performance gains in individual move prediction with as few as 20 games of data. This represents a roughly 250-fold reduction in data requirement compared to prior work (e.g., fine-tuning Maia)

**Additional Comments:**

Here I have some further questions and suggestions:
1. I'd recommend also inviting human players to measure the model's performance. SInce the goal is to play like humans, then the players themself should be one of the golden metrics.
2. I noticed that you chose N=100 in your hyperparameter, which results in 1100 dims of the embedding rows? Do I understand it correctly? If so, how do you choose this hyper-parameter? What will the performance look like if you scale this number? It'd be better if there were statistics about the number of Lichess human game play and the rationale behind this parameter.
3. Another easy baseline could be context-based meta learning, so basically just add personalized experience into the context when training. This sounds like inherently generalizable because it also trains a network that can take in a new player's history. Why choose the embedding-based method + prototype matching instead of this method?

**Audience:**

Yes

**Audience Explanation:**

The paper did a good job of leveraging the meta-learning technique to address the personalized chess policy problem—the high efficiency of the result (that 250x efficiency improvement is impressive), and it'd be pretty useful for both the chess and chess machine learning community. And as far as I know, this is the first literature working on few-shot learning in the chess domain, so I believe some audience may find it interesting.

**Broader Impact Concerns:**

Not available for this work.

**Claims And Evidence:**

No

**Claims Explanation:**

My major concern is that the paper did not do a good job of discussing its relationship with previous work, especially that presented in the meta-learning domain, for instance, [1] also presents a prototype-based meta learning approach for the few-shot learning problem, which has many similarities with the procedures in the Prototype-Informed Initialization section. The embedding gradient descent (equ 7) in the Democratization Step is also not new, considering there are tons of work on optimizing context/low-dimensional parameters in gradient-based meta learning like [2]. The personalization pretraining described in the Enrichment Step is also closely related to the multi-task and meta-learning literature.

Thus, I'd prefer to treat this paper as a nice application paper, rather than something novel or proposed by the authors' claim.


[1] Snell J, Swersky K, Zemel R. Prototypical networks for few-shot learning. Advances in neural information processing systems. 2017;30.
[2] Zintgraf L, Shiarli K, Kurin V, Hofmann K, Whiteson S. Fast context adaptation via meta-learning. InInternational conference on machine learning 2019 May 24 (pp. 7693-7702). PMLR.

**Requested Changes:**

1. A major update on the discussion of the related work. A few words in the related work section is not enough -- especially considering the similarities between those literature. I'd like to see more disucssion on detailed comparison of the proposed methods and exisiting literature, to better clarify what's new in the author's contritbution. I am fine even if the paper is not that new, as long as it is a good application paper. But it is important to properly clarify the contribution.

2. The new LLM experiment in Section 5 is weird -- why suddenly switch to another domain if the title and main content are about chess? It'd be fine if the narrative starts with a general methods that can be applied in different domains. But considering that 95% is about chess, I'd rather move this section to the appendix and only mention a few words about this new exp.

3. Considering this is a meta-learning setting, only comparing the method with one meta-learning variant and one non-meta learning baseline (basically all are Maia's variants) is not enough. There are tons of meta-learning or efficient tuning methods that can also be used as part of your contribution or comparing baselines.

---

> ### Author Response · Authors · 2025-10-15
>
> RC1: We agree that the related work can be more comprehensive. We will revise the paper to clarify our positioning and add a discussion of how our method relates to the meta-learning literature, as detailed below.
>
> Our enrichment step follows multi-task pretraining methodologies common in optimization-based meta-learning, where training across diverse tasks (in our case, prototype players) produces universal parameters for rapid fine-tuning [1,2,3]. The democratization step combines two established techniques. First, context-only optimization: we freeze universal parameters and optimize only low-dimensional player embeddings, following the parameter-efficient adaptation paradigm demonstrated in context-based meta-learning and recent LLMs [4,5,6]. Second, prototype-based initialization: we use a learned prototype matching network to initialize embeddings based on behavioral similarity, following metric-based meta-learning approaches that leverage prototypes for few-shot adaptation [7,8,9,10].
>
> While these individual techniques are well-established, figuring out how to successfully combine several techniques into a pipeline that works surprisingly well for chess is challenging. Such a specific combination for modeling human behavior represents a novel contribution motivated by chess's two-dimensional behavioral structure (strength and style). We require enrichment because population-level embeddings cannot capture individual playing styles within rating bins; we require prototype-based initialization because stylistically similar players can exist across different skill levels, necessitating a learned matching mechanism beyond rating-based lookup; and we require context-only optimization to preserve universal chess knowledge while adapting to each player's coordinates in the behavioral space under low data. To our knowledge, this is the first work that integrates prototype enrichment, learned prototype matching, and context optimization in a unified framework for individual behavior modeling in strategic decision-making domains.
>
> RC2: We agree that the LLM case study serves only as a supporting experiment to show the broader applicability. That’s why we put the results in the discussion section. We would like to weaken any claims about generalization to LLMs in this paper in the revision and move the LLM case study to the appendix, following your suggestion.
>
> RC3: In meta learning settings, the support set and query set are explicitly used differently. In particular, meta learning methods like MAML[1] do not care about the performance on the support set (similar to prototype players) at all; the only goal is to optimize for the query set. While in Maia4All, we consider all players equally, the separation of prototype/unseen players is for the purpose of democratizing the individual behavior modeling process for all players. Therefore, our work is not strictly under a meta-learning setting. Thus, it’s hard to directly compare with classic meta learning baselines. Furthermore, meta learning baselines require splitting the support and query set (prototype and unseen players).  However, the process to separate prototypes and unseen players (prototype selection) is an inherent component in Maia4All. Therefore, it’s challenging to isolate Maia4All completely to apply regular meta learning methods as baselines.

---

> ### Author Response · Authors · 2025-10-15
>
> Additional Comment 1: We would like to reiterate that the Lichess data that we use is exactly real human behavior data. The survey-style human evaluation, such as asking players how well the model imitates them, is meaningful as well. However, we did not have enough resources to conduct a human evaluation yet. We regard this as an important future work.
>
> Additional Comment 2: In Figure 5, we show the effect of the distribution of prototypes (a, b) and prototype quantity per skill level (c, d). As explained at the end of Section 4.2, we chose N=100, considering the trade-off between prototype matching accuracy and player embedding space coverage.
>
> Additional Comment 3: This is an interesting suggestion, which shares some similarity with few-shot prompting and in-context learning in LLMs. However, we notice that history length may vary drastically, from a few to hundreds of thousands of positions, which requires strong long context modeling capabilities that are still challenging in even more mature domains such as language modeling. Furthermore, Maia-2 is a position-wise model, adding “personalized experience” or history during training or inference as contexts involves forward passes for every position in the history, which could potentially cause severe efficiency issues.
>
> [1] Chelsea Finn, Pieter Abbeel, and Sergey Levine. Model-agnostic meta-learning for fast adaptation of deep networks. In *International Conference on Machine Learning (ICML)*, pages 1126–1135, 2017.
>
> [2] Alex Nichol, Joshua Achiam, and John Schulman. On first-order meta-learning algorithms. arXiv preprint arXiv:1803.02999, 2018.
>
> [3] Feiyang Wang, Xin Wang, Shiqi Wang, Zehuan Yuan, Wenwu Zhu, and Peng Cui. LiMAML: Personalization of deep recommender models via meta learning. In *ACM SIGKDD Conference on Knowledge Discovery and Data Mining*, pages 5481–5490, 2024.
>
> [4] Luisa M. Zintgraf, Kyriacos Shiarlis, Vitaly Kurin, Katja Hofmann, and Shimon Whiteson. Fast context adaptation via meta-learning. In *International Conference on Machine Learning (ICML)*, pages 7693–7702, 2019.
>
> [5] Edward J. Hu, Yelong Shen, Phillip Wallis, Zeyuan Allen-Zhu, Yuanzhi Li, Shean Wang, Lu Wang, and Weizhu Chen. LoRA: Low-rank adaptation of large language models. In *International Conference on Learning Representations (ICLR)*, 2022.
>
> [6] Neil Houlsby, Andrei Giurgiu, Stanislaw Jastrzebski, Bruna Morrone, Quentin de Laroussilhe, Andrea Gesmundo, Mona Attariyan, and Sylvain Gelly. Parameter-efficient transfer learning for NLP. In *International Conference on Machine Learning (ICML)*, pages 2790–2799, 2019.
>
> [7]  Jake Snell, Kevin Swersky, and Richard S. Zemel. Prototypical networks for few-shot learning. In *Advances in Neural Information Processing Systems (NeurIPS)*, pages 4077–4087, 2017.
>
> [8] Eleni Triantafillou, Tyler Zhu, Vincent Dumoulin, Pascal Lamblin, Utku Evci, Kelvin Xu, Ross Goroshin, Carles Gelada, Kevin Swersky, Pierre-Antoine Manzagol, and Hugo Larochelle. Meta-dataset: A dataset of datasets for learning to learn from few examples. In *International Conference on Learning Representations (ICLR)*, 2020.
>
> [9] Andrei A. Rusu, Dushyant Rao, Jakub Sygnowski, Oriol Vinyals, Razvan Pascanu, Simon Osindero, and Raia Hadsell. Meta-learning with latent embedding optimization. In *International Conference on Learning Representations (ICLR)*, 2019.
>
> [10] Flood Sung, Yongxin Yang, Li Zhang, Tao Xiang, Philip H.S. Torr, and Timothy M. Hospedales. Learning to compare: Relation network for few-shot learning. In *IEEE Conference on Computer Vision and Pattern Recognition (CVPR)*, pages 1199–1208, 2018.

---

### Review · Reviewer_hY4j · 2025-09-30

**Summary Of Contributions:**

This paper introduces Maia4All, a framework for modeling individual chess-playing behavior with significantly reduced data requirements. Maia4All uses a two-stage approach: (1) an enrichment step that fine-tunes the base model on selected "prototype" players with rich game histories, adapting the model from population-level to individual-level modeling, and (2) a democratization step that uses a Prototype Matching Network to initialize embeddings for new players based on similar prototypes, then fine-tunes embeddings on limited data. The method achieves comparable accuracy gains with 20 games, vs the 5,000 games previously required. The authors include a supplementary case study on adapting LLMs to individual writing styles to demonstrate broader applicability.

Strengths:
While the components are known, the explicit framing of enrichment (population → individual-level parameters) + democratization (prototype-informed initialization) as a systematic approach to the cold-start problem in behavioral modeling is a nice conceptual contribution. Applying this to individual human behavior modeling in chess is an interesting direction. Experiments show a large scale of data efficiency gain (250×) in this specific domain.

Weaknesses:
1. As a non-expert in chess game modeling, the 1.8 percentage point gain (51.4% -> 53.2%) seems small in absolute terms. While the authors argue this is meaningful given human unpredictability, the practical impact for downstream applications (personalized teaching, collaboration) remains unclear.
2. I wonder what happens between 2,500 and 5,000 games? Does Maia4All match or exceed Maia-Individual's performance? This is crucial for understanding whether data efficiency comes at the cost of performance.
3. Prototypes are selected primarily by most frequent players, rather than style diversity. It would be interesting to see an analysis of prototype coverage in behavioral style space.
4. The LLM case study is the only evidence for broader applicability, but the improvements are marginal. Therefore the claim of potential for broader applications is overstated given this limited validation.

**Audience:**

Yes

**Audience Explanation:**

Applying the explicit framework of enrichment + democratization to individual human behavior modeling in chess is an interesting direction.  The authors claim the framework leads to a performance improvement and more efficient use of data, which could be of interest to the broader community.

**Claims And Evidence:**

Yes

**Claims Explanation:**

The authors provide detailed experiment analysis and reported results with baseline comparisons. However there are concerns regarding whether the proposed framework has a significant improvement on the current methods. See weaknesses above.

**Requested Changes:**

Please see weaknesses above.

---

> ### Author Response · Authors · 2025-10-15
>
> W1 and W2: The mentioned 1.8 percentage point gain (51.4% -> 53.2%) represents the improvement with only ~20 games. The baseline method, Maia-Individual, was still achieving negative improvement with 1000 games. In the human move prediction problem for amateur players, the ceiling accuracy is far below 100% given the randomness and diversity of their decisions—even the same player won’t always make the same decision when faced with the same position. The state-of-the-art human-like chess engines still struggle to reach 55% in general. We agree that the mentioned downstream applications were not included in the evaluation, and we regard these as important and immediate future works. On the other hand, data efficiency does not come at the cost of performance. Following the reviewer’s suggestion, we ran Maia4All with ~5000 game histories and achieved 56.2%, an almost 5 percentage point gain from the base model performance of 51.4%. Note that the baseline method, Maia-Individual, improved from 53.2% to 55.0% with a 1.8 percentage point gain using 5000 games per player.  Note that these comparisons are meaningful but have the caveat that there is a test set mismatch caused by the reproducibility issue of Maia-Individual. But we can still observe that Maia4All improved from a worse starting baseline (51.4% vs 53.2%) to a much better resulting performance (56.2% vs 51.4%).
>
> W3: We respectfully disagree with the comment: “Prototypes are selected primarily by most frequent players”. As described in Section 3.2 (Prototype Selection), “the player set is balanced across different skill levels to prevent the model from being biased toward any particular skill level”. To clarify, we first stratify by rating—the first component of variation in chess—and only then take frequent players, so our prototype selection is a first attempt at the prototype coverage that you are asking about. We also presented the effects of prototype diversity (distribution) in Figure 5 (a) and (b), which shows that the default uniform sampling outperforms biased sampling strategies. We agree that a more extensive analysis of this is important future work and will include it in the discussion as such.
>
> W4: We agree that the LLM case study serves only as a supporting experiment to show the broader applicability, and we are aware that the improvement seems to be marginal. However, a small difference in LM loss could potentially induce meaningful downstream language modeling capabilities. That’s why we put the results in the discussion section and mentioned that we need to work on more specific evaluation metrics and more LLM-specific designs to further generalize Maia4All to the language domain. We will weaken our claims about generalization to LLMs in this paper in the revision and potentially move the LLM case study to the appendix.

---

### Decision · Action_Editor_vY1N · 2025-12-14

**Recommendation:** Accept as is

**Audience:**

Yes

**Audience Explanation:**

Yes. Though the domain studied (chess) is narrow, individual behaviour modeling is an area of broad interest to the TMLR audience.

**Claims And Evidence:**

Yes

**Claims Explanation:**

The reviewers expressed no concerns about the claims made in the submission. The primary claims are all supported by well-designed experiments. The only concern, which multiple reviewers brought up, is that the proposed method offers only marginal improvement over prior methods, and the authors should make sure not to overstate their results.